# Physicochemical Properties and Evaluation of Antioxidant Potential of Sugar Beet Pulp—Preliminary Analysis for Further Use (Future Prospects)

**Andrzej Baryga [1], Rafał Ziobro [2,*], Dorota Gumul [2], Justyna Rosicka-Kaczmarek [3] and Karolina Miśkiewicz [3]**

[1]   Faculty of Biotechnology and Food Sciences, Department of Sugar Industry and Food Safety Management, Lodz University of Technology, ul. Wólczańska 171/173, 90-530 Łódź, Poland

[2]   Department of Carbohydrate Technology and Cereal Processing, Faculty of Food Technology, University of Agriculture in Krakow, al. Mickiewicza 21, 31-120 Krakow, Poland

[3]   Faculty of Biotechnology and Food Sciences, Institute of Food Technology and Analysis, Lodz University of Technology, ul Stefanowskiego 2/22, 90-537 Łódź, Poland

*   Correspondence: rafal.ziobro@urk.edu.pl

**Abstract:** High content of pro-health constituents in fruit and vegetable pomaces has led to their utilization as raw materials in food production. They are used mostly in dried form, which is microbiologically stable and allows their storage throughout a longer period. Nevertheless, some materials of these kind are still undervalued, among them sugar beet pulp, which is produced during sugar production in large quantities, often posing an environmental threat, and has been traditionally used for feeding animals. Earlier studies on chemical composition suggested that sugar beet pulp could be highly valuable in terms of health-promoting aspects. Therefore, in this work, research was directed to prove the nutritional potential of this raw material. Thus, an attempt was made to characterize sugar beet pulp in terms of its nutritional and carbohydrate profile, as well as its health-promoting qualities, with particular emphasis on the effect of the extraction on the content of polyphenols and phenolic acids, flavonoids, flavonols, and also their antioxidant activity, measured by ABTS and FRAP methods. The soluble and insoluble fraction of dietary fiber and total dietary fiber were also determined in the pulp. It was found that sugar beet pulp is a valuable source of nutrients (around 10% protein, 7% fat, 8% sugar, 4% ash), dietary fiber (nearly 70%), and has significant amounts of sugars present as free saccharides (fructose and glucose) and polysaccharide residues (arabinose, galacturonic acid, rhamnose, and glucose). In addition, it is a source of polyphenols, flavonoids, and phenolic acids and has a high health-promoting potential regardless of the applied extraction method. Therefore, we may suggest that sugar beet pulp could become an ingredient for pro-health functional food.

**Keywords:** antioxidant activity; nutritional value; polyphenols; sugar beet pulp; sugar profile

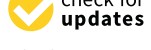



## 1. Introduction

The agricultural industry, through the processing and treatment of raw materials of plant and animal origin, contributes to the formation of large quantities of organic residues. These are both solid and liquid materials, which usually have no further use in the production chain. If they are not properly treated, they can increase the pollution of soil, surface water and groundwater, causing huge problems for the environment [1]. Beet pulp is among such wastes in the sugar manufacture process. According to The Food and Agricultural Organization (FAO), global sugar beet production was over 270 million tons in 2021 [2]. The EU produces the most sugar beet, followed by Asia and North America. More than 113 million tons of sugar beet were produced in EU countries in 2021 [2]. The area under sugar beet cultivation covered around 4.4 million ha worldwide, including nearly 1.5 million ha in EU countries. The use of sugar beet has for many years been directed

solely towards the manufacture of sugar, i.e., sucrose. The by-products resulting from this production process, i.e., pulp and carbonation mud, are organic residues that are difficult to dispose of, with the exception of molasses. Molasses is a valuable by-product used to produce spirit (ethanol), baker's yeast and fodder yeast, glycerin, butanol, acetone, and organic acids (lactic, citric, glutamic). Molasses is also used in the fermentation industry and to refine (enrich with nutrients) pulp for animal feed. In addition, farmers use molasses as an addition to silage to enrich it with carbohydrates. High sugar production in EU countries generates a lot of by-products in sugar production especially Sugar Beet Pulp (SBP), which is obtained after hot extraction of juice from sugar beet cossettes during sugar manufacture [1]. Only in the EU, about 14 million tons (dry substance) of SBP is produced every year [3]. According to Vučurović and Razmovski [4], SBP contains polysaccharides, composed of cellulose, hemicelluloses, and pectins. According to Zheng et al. [5] and Kelly [6], 25 to 36% of hemicelluloses, 20 to 25% of cellulose from 1 to 3% of lignins, and 20 to 25% of pectins were determined in the dry weight of the pulp.

SBP, which is primarily utilized as animal feed, is also subjected to chemical conversion during the production of ethanol. Moreover, there have been attempts to utilize SBP as a source of energy feedstock, specifically for the production of biofuels [7–9].The potential of SBP as a raw material for the biosynthesis of lactic acid is also being considered [10,11]. Another series of trials involved the manufacture of thermoplastic films utilizing SBP as the raw material, which were produced through the use of plasticizers in a twin-screw extruder [12,13]. The resulting composite was described as microfibrils of cellulose dispersed in a pectin matrix. Furthermore, SBP was employed as a source of polyols in the manufacture of urethanes [14]. When combined with polylactic acid, sugar beet pulp yielded polymer composites that exhibited tensile properties comparable to those of conventional plastics [15].

Taking into account the above presented applications of SBP, it should be clearly stated that they do not take into account the pro-health character of this heterogeneous plant material, constituting a specific nutritional matrix, which is partly due to lack of knowledge on this subject. Pulp is a valuable source of dietary fiber, which has begun to receive attention in recent years. The main polysaccharides included in the dietary fiber present in sugar beet pulp are alkali-soluble polysaccharides (ASP), consisting of arabinans, arabinogalactans, and galactans attached as side chains of pectin, and partly bound to cellulose. However, an important aspect has been overlooked concerning the presence in this material of many valuable bioactive compounds from the polyphenol group, which are partially bound with fiber. They include hydroxycinnamic acids (e.g., ferulic acid) attached to arabinogalactans present in sugar beet [16], in a similar manner that they are linked with wheat arabinoxylans [17,18]. In general, ferulic acid derivatives are the group of polyphenols that are most frequently linked with the fiber by covalent (ester, ether of carbon-carbon) bonds [19]. In contrast, the connections between proanthocyanidins and the fiber are usually formed through hydrogen bonds, a hydrophobic effect, van der Waals forces, or other non-covalent interactions [20]. According to Liu et al. [20], polyphenols and dietary fiber interact with each other through the creation of organized bonds, which are fortified by non-covalent interactions such as electrostatic forces, ionic bonding, hydrogen bonding, and the hydrophobic effect. These interactions are typically a result of physical damage and aging of plant tissues. Thanks to their existence, SBP constituents can also be used as bioactive food ingredients and bioproducts. Aside from their bioactive properties, polysaccharides also have a substantial functional impact. In the case of sugar beet pectin, its high degree of acetylation and feruloylation, as well as the presence of neutral sugar side chains, prevent it from gelling. This is in contrast to commercially available pectins, which are mainly extracted from citrus [21]. The potential of sugar beet pectin as an emulsifier for beverages has been extensively studied. Its emulsion stabilization properties are attributed to the significant levels of acetyl and protein present in sugar beet pectin [21]. The compounds mentioned above have been found to possess various beneficial effects, such as hypoglycemic and hypocholesterolemic properties, as

well as the ability to combat cancer, inflammation, allergies, and eliminate viruses and bacteria. Additionally, they demonstrate prebiotic and anti-adhesive properties against food pathogens, reduce postprandial glycemia and hypertension, and lower the risk of developing cardiovascular diseases, cataracts, diabetes, genetic damage, degenerative bone changes, and neurodegenerative conditions such as Alzheimer's disease [22–37]. It is therefore important to analyze the chemical composition of beet pulp with particular regard to polyphenols as compounds integrally linked with non-starch polysaccharides present in the sugar beet pulp. It is crucial to consider this issue because it would enable the possibility of using pulp in the food industry, after previous microbiological stabilization of fresh pulp, for example, through freeze-drying. Freeze-drying, besides the inhibition of thermal degradation processes and microbiological spoilage, will also guarantee a high degree of preservation of the natural chemical composition of the obtained pulp, so it is possible to propose the use of such a post-production raw material in the creation of innovative food products [38,39].

Freeze-drying has not been previously investigated for SBP, although it is widely recognized as a minimally invasive technique with regards to nutrient preservation [38,39]. Therefore, the analysis of SBP after freeze-drying can be considered reasonable in order to explore potential novel applications of this by-product in food technology.

A comprehensive characterization of beet pulp as a source of health-promoting compounds must involve the effect of solvents on the extraction efficiency of polyphenols from plant material. According to many authors [40–44], a lot of factors can influence the final result of the determination of the polyphenolic compounds concentration in the sample. These include the type of method for the determination of polyphenols, extraction conditions, i.e., type of extraction reagent, composition of extraction solvent, temperature, extraction time, proportionality of solvent to solids, and product storage conditions, among others.

The effect of the solvent used on the level of phenolic compounds determined in the material was analyzed in scientific studies on various plant material in which the extractants were organic solvents, but little attention was given to fruit and vegetable pomace, especially SBP. In the study of Mokrani and Madani [44] on the effect of the type of solvent used to extract polyphenols and flavonoids from peach fruit, 60% acetone was found to be the best. Similarly, in the study on eggplant peel, it was proved that 70% methanol was the most efficient for extraction of anthocyanins, while the best extractant for total polyphenols, flavonoids, and tannins was 70% acetone [42]. According to Sulaiman et al. [45] who extracted polyphenols from vegetables, 70% acetone was the most favorable solvent, and according to Do et al. [46], 100% ethanol was the best solvent to isolate polyphenols from the medicinal spice *Limnophilia aromatica*. A comparison was made between Soxhlet extraction and maceration methods from the whole plant of *Osbeckia parvifolia*. Spectrophotometric techniques were employed to assess the levels of total phenolics, tannins, and flavonoids present in the extracts. It was shown that the whole plant has a high content of polyphenols and flavonoids and tannins. Results of antioxidant evaluation showed that methanol (both macerated and Soxhlet) extracts displayed stronger antioxidant properties compared to other extracts. The methanol extract obtained through a Soxhlet apparatus exhibited protective effects against protein denaturation and the degradation of erythrocyte membranes [43]. According to the study of Su et al. [47], among the solvents used for extraction of litchi pulp, aqueous acetone proved to be the most efficient in extracting total free phenolic compounds, while a blend of methanol, ethanol, and ethyl acetate followed in effectiveness. Furthermore, it was demonstrated that the acid hydrolysis method resulted in a greater release of bound phenolic compounds compared to the alkaline hydrolysis method [47]. The effect of solvent was also studied for freeze-dried berries such as blackberries, black mulberries, and strawberries. Extraction was performed using three distinct organic solvents (methanol, ethanol, and acetone), as well as distilled water. Different concentrations of these solvents, 70, 50, and 100%, were also used along with acetic acid. The solutions obtained from the extraction process were employed to evaluate the total

phenolic and anthocyanin content. Methanol was identified as the most efficient organic solvent for extracting antioxidants, followed by water, ethanol, and acetone. This may be due to better solvation, i.e., ion interaction between the solvent and the ions of the substance. Among the different solvent concentrations used, extraction with 70% acetone was the most efficient for polyphenols from white mulberry and blackberry. For strawberry, high polyphenol values were obtained with extraction in 50 and 70% acetone. The lowest value of total polyphenols for all fruits was obtained with an acetic acidified acetone mixture [48]. While the above-mentioned studies used various types of plant tissues (fresh, air-dried), the effect of the extractant on the amount of phenolic compounds from freeze-dried SBP was not analyzed, so its examination would be a novelty.

This study aimed to investigate the impact of various extraction solvents (water and 80% aqueous solutions of methanol, ethanol, and acetone) on the content of phenolic compounds such as total polyphenols, flavonoids, flavonols, and phenolic acids, as well as to estimate the antioxidant potential of sugar beet pulp after freeze-drying. In addition, due to the nature of the analyzed samples, the aim was also to determine the chemical composition, the amounts of soluble and insoluble fractions of dietary fiber, total fiber, and the characteristics of the non-starch polysaccharide fraction of this valuable biological material, including the carbohydrate profile, and galacturonic acid content and structure. The obtained results will allow identification of the possible directions of future use of this valuable by-product generated in sugar beet processing.

## 2. Materials and Methods

### 2.1. Materials

The study material consisted of sugar beet pulp obtained from an agricultural farm (Kruszwica, Piecki, Poland) which was freeze-dried ($-47$ °C, 37 Pa, 24 h; FreeZone 6, Labconoco, Kansas city, MO, USA) and ground by a laboratory grinder (Grindomix, GM 200, Retsch GmbH, Hahn, Germany) at 7000 rpm in two consecutive runs, each lasting for 5 s.

Sugar beets were grown conventionally in the fields with good nutrient abundance and low levels of acidic soils. The soil type observed in the cultivation area was mainly black soil (chernozem) and brown earth (parent Quaternary clayey formations), belonging mostly to the second and third soil quality classes. In August, the fields were fertilized with manure (34 t/ha) and immediately plowed up to 4 h after the fertilization process. Subsequently, after 2–4 days, the rolling of the subsoil was carried out. In mid-October, the soil was subjected to a winter plowing process at a depth of about 27 cm. During the spring period, potassium salt (60% KCl at a dose of 100 kg/ha) in the form of potassium oxide (40% $K_2O$), magnesium (6% MgO), sulfur (12.5% $SO_3$), and sodium (4% $Na_2O$) were applied together with a mineral fertilizer containing nitrogen (18%), phosphorus in the form of di-ammonium phosphate (46% $P_2O_5$), and potassium (10%). The dosage of the fertilizer was adjusted to the plants' needs and reached 130 kg/ha. The fertilization process was followed by blending the fertilizers with a cultivator at a depth of 8 cm. This work was followed by spreading highly concentrated nitrogen fertilizer for pre-sowing application (late March) at a rate of 140 l/ha and post-sowing application (early June). After a few days, the sowing of beets began (from late March to the first decade of April).

Vegetation of sugar beets lasted 180 days. Root harvesting was conducted mechanically, and beets were stored and transported to the sugar factory for production.

### 2.2. Determination of Nutritional Compounds in Freeze-Dried Sugar Beet Pulp

2.2.1. Determination of Protein, Fat, Ash, Carbohydrates, Reducing Sugars, and Dietary Fiber

The content of essential nutrients in the analyzed sample was determined by AOAC methods [49]. Protein content (N $\times$ 5.7) was assessed by Kjeldahl's method (AOAC 920.87), applying extraction unit Kjeltec 2200 (Foss, Denmark), total carbohydrate content was evaluated following AOAC 974.06, fat according to Soxhlet's method AOAC 953.38, applying

the extractor Soxtec Avanti 2055 (Foss, Denmark), and ash by gravimetric method (AOAC 930.05). The content of non-starch polysaccharides, i.e., total, soluble, and insoluble dietary fiber, was determined by the method 32-07 while reducing sugars were determined according to the method 906.01 AOAC [49]. Each of the above determinations was performed in at least 2 replicates.

### 2.2.2. Qualitative and Quantitative Determination of Free Carbohydrate Content in Sugar Beet Pulp by HPLC

The content of free carbohydrates in sugar beet pulp was determined using high performance liquid chromatography HPLC (Thermo Fisher Scientific Inc., Waltham, MA, USA) by means of Shodex NH2P-50 series columns with own modification.

To determine the carbohydrate content, 2 g samples were weighed with an accuracy of 0.0001 g and placed into 50 mL centrifuge tubes. Next, 20 mL of ultrapure water was added, and the tubes were shaken in a water bath at 85 °C for 60 min. After centrifuging at $3024 \times g$ for 20 min at 20 °C, the supernatants were decanted and deproteinized with Carrez I and II reagents. The samples were then centrifuged again at $3024 \times g$ for 10 min at 20 °C, and the supernatants were quickly decanted and filtered using a nylon syringe filter with a 0.2 μm pore size. The filtrates were analyzed for free sugar content using a UHPLC+ Dionex UltiMate 3000 system equipped with a refractive index detector and an Asahipak NH2P-50 4E column. Isocratic elution was carried out with 70/30 (*v/v*) acetonitrile/water as the mobile phase. The flow rate was set to 1.0 mL/min, and the column temperature was 30 °C. The identification of glucose, fructose, sucrose, trehalose, kestose, and rafinose was conducted by comparing their retention times with authentic standards (Sigma-Aldrich, Saint Louis, MO, USA). The quantification was carried out using an external standard. The samples were analyzed in triplicate.

### 2.2.3. Qualitative and Quantitative Determination of Carbohydrate Content in Sugar Beet Pulp after Hydrolysis by HPLC

The content of carbohydrates in SBP after hydrolysis was determined by HPLC (Thermo Fisher Scientific Inc., Waltham, MA, USA) according to the method described by Hosseini et al. [50] and Gruska et al. [51] with own modification.

To determine the carbohydrate content, 0.010 g samples were weighed into 5 mL tubes with an accuracy of 0.0001 g; 5 mL of 2M trifluoroacetic acid was added, and the tubes were inserted into a shaking water bath at 100 °C for 150 min. The samples were then centrifuged ($3024 \times g$, 20 min, 20 °C). Approximately 2.0 mL of supernatants were then taken, transferred to test tubes, and dried under a stream of nitrogen to evaporate the trifluoroacetic acid. The dry residue was dissolved in 1 mL of ultrapure water. The solutions were filtered through 0.2 μm nylon syringe filters into the autosampler vials. The filtrates were analyzed for the content of carbohydrates using a UHPLC+ Dionex UltiMate 3000 system (Thermo Fisher Scientific Inc., Waltham, MA, USA) equipped with a refractive index detector (Shimadzu, Kioto, Japan) and a Rezex RPM Monosaccharide Pb$^{2+}$ (7.8 × 300 mm, 8.0 μm particle size; Phenomenex, USA). Isocratic elution was carried out with ultrapure water as the mobile phase. The flow rate was set to 0.6 mL/min, and the column temperature was 80 °C. Glucose, xylose, celobiose, rhamnose, galactose, arabinose, and mannose were identified by comparing their retention times with respective standards (Sigma-Aldrich, Saint Louis, MI, USA). Quantification was performed using an external standard. All measurements were performed in triplicate.

### 2.2.4. Determination of Galacturonic Acid Content in Sugar Beet Pulp

Galacturonic acid in sugar beet pulp was determined by the colorimetric method with 3,5-dimethylphenol according to the methodology described in the Polish standard PN-A-75113 (1997) [52].

2.2.5. The Structure of Polysaccharides Present in Sugar Beet Pulp

The structure of polysaccharides present in sugar beet pulp was determined according to the method described by Kazemi et al. [53].

The share of the homogalacturonate domain (HG) in sugar beet pulp was determined using the formula:

$$HG \ (g/100 \ g) = GalA \ (g/100 \ g) - Rha \ (g/100 \ g)$$

where:

HG—content of homogalacturonate in sugar beet pulp (g/100 g);
GalA—content of galacturonic acid in sugar beet pulp (g/100 g);
Rha—content of rhamnose in sugar beet pulp (g/100 g).

The share of the rhamnogalacturonan domain (RG-I) in sugar beet pulp was calculated using the following formula:

$$RG\text{-}I \ (g/100 \ g) = [[GalA \ (g/100 \ g) - HG \ (g/100 \ g)] + Rha \ (g/100 \ g) + Gal(g/100 \ g) + Ara \ (g/100 \ g)]$$

where:

RG-I—rhamnogalacturonate content in sugar beet pulp (g/100 g);
GalA—content of galacturonic acid in sugar beet pulp (g/100 g);
Gal—galactose content in sugar beet pulp (g/100 g);
Rha—rhamnose content in sugar beet pulp (g/100 g);
Ara—arabinose content in sugar beet pulp (g/100 g).

The share of the linear fraction in polysaccharides (MR1) present in sugar beet pulp was calculated using the following formula:

$$MR1 = GalA/(Rha + Ara + Gal + Xyl + Fru)$$

where:

GalA—content of galacturonic acid in sugar beet pulp (g/100 g);
Gal—galactose content in sugar beet pulp (g/100 g);
Rha—rhamnose content in sugar beet pulp (g/100 g);
Ara—arabinose content in sugar beet pulp (g/100 g);
Xyl—xylose content in sugar beet pulp (g/100 g);
Fru—fructose content in sugar beet pulp (g/100 g).

The share of the rhamnogalacturonan (MR2) domain in polysaccharides present in sugar beet pulp was determined on the basis of the ratio:

$$MR2 = Rha \ (g/100 \ g)/GalA \ (g/100 \ g)$$

where:

GalA—content of galacturonic acid in sugar beet pulp (g/100 g);
Rha—rhamnose content in sugar beet pulp (g/100 g).

The branching length of the side chains attached to rhamnogalacturonan (MR3) in polysaccharides present in sugar beet pulp was determined from the following relationship:

$$MR3 = Gal(g/100 \ g) + Ara(g/100 \ g))/Rha \ (g/100 \ g)$$

where:

Gal—galactose content in sugar beet pulp (g/100 g);
Rha—rhamnose content in sugar beet pulp (g/100 g);
Ara—arabinose content in sugar beet pulp (g/100 g).

*2.3. Determination of Bioactive Compounds (Polyphenols)*

2.3.1. Preparation of Extracts

Extraction: (reagent-water and 80% aqueous solution of methanol, acetone, and ethanol). Approximately 0.6 g of the test material was extracted using 30 mL of 80% ethanol/methanol/acetone or water, at an ambient temperature (20 °C) for 2 h in a covered water bath with a shaker (Memmert, WB 22, Schwabach, Germany). The extract was then centrifuged in a bench-top centrifuge (MPW-350, Warsaw, Poland) at 4500 rpm for 15 min ($1050\times g$). The whole sample was stored in the freezer ($-20$ °C) for further analysis.

2.3.2. Determination of Total Polyphenol Content (TPC) by Folin–Ciocalteau Reagent (FCR)

The determination of total polyphenol content (TPC) using the Folin–Ciocalteau reagent (FCR) was conducted following Singleton et al. [54]. Initially, 5 mL of the extract was diluted with distilled water to a final volume of 50 mL. Subsequently, 5 mL of the diluted extract was mixed with 0.25 mL of the FCR (previously diluted with distilled water in a 1:1 $v/v$ ratio) and 0.5 mL of 7% $Na_2CO_3$. The contents were shaken using a Vortex WF2 (Janke & Kunkel, Staufen, Germany) and left in the dark for 30 min. After this time, the absorbance was measured using a spectrophotometer Helios Gamma 100–240 (Runcorn, UK), at the wavelength $\lambda = 760$ nm. The results were converted to mg catechin per 100 g d.m. (mg CE/100 g d.m.).

2.3.3. Determination of Total Polyphenol Content (TPC without FCR), Phenolic Acids, Flavonols

Total polyphenol content (TPC without FCR), phenolic acids, and flavonols were determined according to the spectrophotometric method described by Mazza et al. [55] with modification by Oomah et al. [56].

A volume of 0.1 mL of extract was taken into a test tube, and 2.4 mL of 2% HCl in 75% ethanol was added. The contents were shaken using a Vortex WF2 (Janke & Kunkel, Staufen, Germany), and the absorbance was measured in a spectrophotometer Helios Gamma 100–240 (Runcorn, UK), at the wavelengths $\lambda = 280$ nm (TPC without FCR), $\lambda = 320$ nm (phenolic acids), and $\lambda = 360$ nm (flavonols). TPC was expressed in mg of catechin per 100 g d.m., phenolic acids in mg of ferulic acid per 100 g d.m. (mg FAE/100 g d.m.), flavonols in mg quercetin equivalent per 100 g d.m. (mg QE/100 g d.m.) [55,56].

2.3.4. Determination of Flavonoid Content

Flavonoid content was determined according to the method of Andary [57] as described by El Hariri et al. [58]. A volume of 0.5 mL of extract was taken into a test tube, and 1.8 mL of distilled water and 0.2 mL of 2-aminoethyldiphenylborate reagent were added. The contents were shaken using a Vortex WF2 (Janke & Kunkel, Staufen, Germany), and the absorbance was measured in a spectrophotometer Helios Gamma 100–240 (Runcorn, UK), at the wavelength $\lambda = 404$ nm [58]. Flavonoid content was expressed as mg of rutin per 100 g d.m. (mg RE/100 g d.m.)

2.3.5. Antioxidant Activity by Using ABTS

The antioxidant activity of the samples was evaluated using the ABTS (2,2′-azino-bis(3-ethylobenzothiazoline-6-sulphonic acid)-diammonium salt) analytical method, as described by Re et al. [59]. The ABTS stock solution was dissolved in water to achieve a concentration of 7 mM. The ABTS radical cation (ABTS$^{+\bullet}$) was generated by reacting the ABTS stock solution with 2.45 mM potassium persulfate (final concentration) and allowing the mixture to stand in the dark at room temperature for 12–16 h before use. The rate of ABTS$^{+\bullet}$ radical scavenging in the presence of the sample was determined at 734 nm using a Helios Gamma 100–240 spectrophotometer (Runcorn, UK). The ABTS$^{+\bullet}$ solution was diluted in a PBS buffer (pH 7.4) to obtain an absorbance value of $0.700 \pm 0.05$ for analysis of extracts. A volume of 2.00 mL of ABTS$^{+\bullet}$ solution and corresponding extract

in the PBS buffer were used. Radical scavenging activity was monitored at 37 °C, and the decolorization after 6 min was used to determine the antioxidant activity, expressed as Trolox Equivalents Antioxidant Capacity (mM of Trolox per kg of dry matter sample). A calibration curve was prepared using Trolox solutions in the concentration range of 0–2.5 mM ($R^2$ = 0.9957).

### 2.3.6. Determination of Reducing Power Potential by the Method of FRAP Assay

The methodology of Benzie and Strain [60] was employed to evaluate the ferric reducing ability of sugar beet extracts. To prepare the FRAP reagent, 300 mM acetate buffer (pH 3.6), 40 mM 2,4,6-tri(2-pyridyl)-triazine (TPTZ) (dissolved in 40 mM HCl), and 20 mM ferric chloride (dissolved in water) were mixed in a ratio of 10:1:1 (*v/v*), respectively. The extract (0.4 mL) was mixed with 1.2 mL of the FRAP reagent and 2.0 mL of ultrapure water. After 4 min of incubation in the dark, the absorbance was read at 593 nm using a Shimadzu Spectrophotometer UV-Visible 1800 (Japan). The results were expressed as mmol of $Fe^{2+}$ per kilogram dry mass of the sample, with reference to a dose–response curve for iron (II) sulfate.

### 2.4. Statistical Analysis

All analyses were carried out at least in duplicate. The results were expressed as means and standard deviations (SD). The experimental data were subjected to analysis of variance (Duncan's test), at the confidence level of 0.05, by the use of software Statistica v. 8.0 (Statsoft, Inc., Tulsa, OK, USA). Correlation coefficients were calculated with the use of Statistica 8.0 PL.

## 3. Results and Discussion

Sugar beet pulp is the residue remaining after the extraction of juice from beet cossettes. It is a production waste consisting mainly of fiber components as well as protein, fat, and ash. The content of protein, fat, and ash in the analyzed pulp was, respectively, 10.11, 6.90, and 4.16 g/100 g d.m. (Table 1). In addition, other components such as total sugars and reducing sugars were determined in amounts of 8.27 and 6.92 g/100 g d.m.

**Table 1.** Characteristics of freeze-dried sugar beet pulp.

| Constituents | | Content (g/100 g d.m.) |
|---|---|---|
| Protein | | $10.11 \pm 0.12$ |
| Fat | | $6.90 \pm 0.01$ |
| Reducing sugars | | $6.92 \pm 0.01$ |
| Total sugars | | $8.27 \pm 0.02$ |
| Ash | | $4.16 \pm 0.04$ |
| Dietary fiber | Soluble (SDF) | $17.83 \pm 0.15$ |
| | Insoluble (IDF) | $49.77 \pm 0.01$ |
| | Total (TDF) | $67.60 \pm 0.11$ |

Data are presented as mean $\pm$ SD (n = 3; $p \leq 0.05$).

Mohdaly et al. [40] determined the fat level of 6.92 g/100 g d.m., protein 10.8 g/100 g d.m., and ash oscillating around 6.64 g/100 g d.m. in sugar beet pulp, while according to Dinand et al. [61], protein content equaled 7 g/100 g d.m, fat–2 g/100 g d.m., and ash varied from 4.5 to 5 g/100 g d.m. Asadi [1] recorded the protein content of beet pulp at 7 g per 100 g of dry matter, ash at 5 g, and fat at 0.5 g. It could be said that the values obtained in the current study are comparable to the results of other authors, and some differences are due to the sugar beet variety, growing region, and soil, climatic and agrotechnical conditions [1].

Sugar beet pulp was analyzed to determine the content of free sugars. It was found that fructose was present in the largest amount, the content of which was at the level of 1.12 g/100 g (Table 2). Glucose (0.84 g/100 g) and a relatively small amount of sucrose (0.10 g/100 g) were also found in the obtained sugar beet pulp. Among the free sugars, the presence of raffinose, trehalose, and kestose, i.e., sugars typical of sugar beet, were

not detected. It is commonly known that those sugars are transferred to beet juice and processed together with sucrose (inhibiting its crystallization). Therefore, they are not present in sugar beet pulp.

**Table 2.** Profile of free sugars and composition and structure of polysaccharides of freeze-dried sugar beet pulp.

| Free Sugars Composition (g/100 g) | |
|---|---|
| Fructose | $1.12 \pm 0.02$ |
| Glucose | $0.84 \pm 0.07$ |
| Sucrose | $0.10 \pm 0.00$ |
| Rafinose | nd |
| Trehalose | nd |
| Kestose | nd |
| **Carbohydrate Composition (g/100 g)** | |
| Celobiose | $0.29 \pm 0.03$ |
| Glucose | $3.25 \pm 0.29$ |
| Xylose | $0.17 \pm 0.02$ |
| Rhamnose | $3.35 \pm 0.21$ |
| Arabinose | $19.36 \pm 0.72$ |
| Mannose | $1.01 \pm 0.09$ |
| Galacturonic acid | $4.80 \pm 0.09$ |
| **Sugar Molar Ratios** | |
| HG (%) | 1.45 |
| RG-I (%) | 26.06 |
| MR1 | 0.21 |
| MR2 | 0.7 |
| MR3 | 5.78 |

Data are presented as mean $\pm$ SD (n = 3; $p \leq 0.05$); nd—not detected; HG—homogalacturonan; RG-I—rhamnogalacturonan I; MR1—the share of the linear fraction in polysaccharides present in sugar beet pulp; MR2—the share of the rhamnogalacturonan domain in polysaccharides present in sugar beet pulp; MR3—the branching length of the side chains attached to rhamnogalacturonan in polysaccharides present in sugar beet pulp.

The polysaccharide fraction of sugar beet pulp includes non-starch polysaccharides, i.e., cellulose, as evidenced by the presence of glucose in the carbohydrate profile obtained for the tested pulp, and relatively small amounts of cellobiose as a product of incomplete cellulose hydrolysis. Another group of polysaccharides identified in the tested sugar beet pulp are hemicelluloses. Their presence in the carbohydrate profile is indicated by arabinose and small amounts of xylose and mannose (Table 2). On this basis, it is clear that the main group of hemicelluloses in the tested sugar beet pulp are arabinans [62]. The identified galacturonic acid and rhamnose (Table 2) in the tested sugar beet pulp indicate the presence of pectins among the non-starch polysaccharides. Thus, it can be said that the results we obtained confirm the results of earlier authors [15,61,63,64]. Beet pulp is a by-product of sugar beet processing with a high content of structural polysaccharides, i.e., cellulose (22–30% d.m.), hemicellulose (22–30% d.m.), pectin (24–32% d.m.), and lignin (1–8% d.m.) [63,64]. According to Dinand et al. [61], the levels of polysaccharides vary between 26 and 35% for pectins, 22 and 33% for cellulose, and 3.5 and 6.5% for lignin. Fingenstadt [15] reported the fiber constituents to be mostly hemicellulose (28%), cellulose (19%), and pectin (18%).

On the basis of the identified sugars included in the polysaccharides of the sugar beet cell wall, the shares of individual domains, which inform about the structure of these polysaccharides, were counted. The proportion of the homogalacturonate (HG) domain was calculated based on the content of galacturonic acid and rhamnose. Galacturonic acid is the main component of the HG domain, and the short backbone of the HG domain in sugar beet pulp was illustrated by its low galacturonic acid content (1.45%) as shown in Table 2. The side chains of the RG-I domain contain neutral sugars such as galactose and arabinose, and the predominance of arabinose and absence of galactose indicate that the arabinan side chains are abundant. The high content of neutral sugars in SBP suggests that

the RG-I region is highly branched, which is further confirmed by the high molar ratios of (Gal + Ara)/Rha (5.78) (MR3). These ratios are often used as indicators of the extent of branching in the RG-I domain. The obtained MR3 value (5.78) compared to the share of the linear fraction in polysaccharides (MR1 = 0.21) and the share of the RG-I domain alone (MR2 = 0.70) clearly indicates a relatively high branching of the side chains of cell wall polysaccharides present in SBP. Knowledge of the structure of polysaccharides is important for determining the potential directions of use of the tested material, i.e., sugar beet pulp. To ensure optimal use of resources, it is crucial to effectively utilize the sugars that are present in notable quantities (namely glucose, arabinose, rhamnose, and galacturonic acid) in sugar beet pulp to produce valuable products.

One of the possible solutions to exploit the potential of these monosaccharides would be their fractionation and then individual processing to obtain interesting products [62]. Glucose can be used by various microorganisms in the reaction to obtain organic acids and alcohols [65,66]. It was shown that the polysaccharide fraction of the tested beet pulp is rich in arabinose. Arabinose is considered a valuable product owing to its ability to lower blood glucose levels and insulin response when taken along with a diet containing high levels of sugar and starch [67]. An example of a beneficial transformation of arabinose is arabitol, obtained by hydrogenation of arabinose or its fermentation reaction with yeast [68,69]. L-arabitol has the potential to be a sweetener with a low glycemic index and a prebiotic effect, making it an interesting candidate for further investigation [69].

All of the polysaccharides described above are structural components that make up the cell walls of sugar beets that are part of dietary fiber. It should be remembered that dietary fiber is a chemically heterogeneous multicomponent complex, in which we distinguish insoluble and soluble fractions with different physiological effects on the human body. Insoluble fiber fraction is recommended for prevention and treatment of colon diseases (habitual constipation, irritable bowel syndrome, hemorrhoids, and diverticulosis). The soluble fraction of dietary fiber has hypocholesterolemic, hypoglycemic, and anticancerogenic effects [70,71].

Beet pulp contained insoluble fractions of dietary fiber in the amount of 49.7 g/100 g d.m., soluble fraction—17.83 g/100 g d.m., and total dietary fiber 67.60 g/100 g d.m. (Table 1).

The polysaccharides present in the tested sugar beet pulp have relatively high added value. They have a beneficial effect on human health [72]. The influence of sugar beet pulp (SBP) on cholesterols level was determined to be beneficial [73]. The use of sugar beet fiber for food processing is limited due to its texture and taste, but it has already been used as a thickening or bulking agent in food technology [74].

In summary, it can be said that SBP is a valuable source of dietary fiber. However, an important aspect concerning the presence of many valuable bioactive compounds from the polyphenol group, which are partially bound to dietary fiber, has been omitted.

Discussing polyphenols in SBP, one should consider the influence of extractants on the levels of total polyphenols, flavonoids of flavonols, and phenolic acids and estimated antioxidant activity.

Table 3 shows the results of total polyphenol content in beet pulp using different extractants (80% methanol, 80% ethanol, 80% acetone, and water). It could be observed that TPC determined after the extraction with 80% methanol or 80% acetone was at the same level, significantly higher than in the case of ethanol extraction. Thus, it can be expected that the use of organic solvents, namely, 80% methanol or 80% acetone contributed to the comparable extraction of polyphenols from the sample, except for the extraction with 80% ethanol.

**Table 3.** The content of total polyphenols and flavonoids in extracts of freeze-dried sugar beet pulp depending on the type of solvent used in the extraction process.

| Type of Polyphenol Extraction from SBP | TPC with FCR (mg CE/100 g d.m.) | TPC without FCR (mg CE/100 g d.m.) | Content of Flavonoids (mg RE/100 g d.m.) |
|---|---|---|---|
| methanol extraction | 366.00 ± 5.77 [b,*] | 338.67 ± 2.51 [c] | 370.00 ± 5.65 [d] |
| ethanol extraction | 121.45 ± 5.03 [a] | 48.73 ± 3.21 [b] | 56.2 ± 1.27 [a] |
| acetone extraction | 381.06 ± 10.81 [b] | 15.32 ± 0.91 [a] | 173.3 ± 3.09 [b] |
| water extraction | 520.02 ± 15.82 [c] | 458.33 ± 5.71 [d] | 304.00 ± 5.19 [c] |

* Different letters indicate significant differences between individual samples ($p \leq 0.05$); data are presented as mean ± SD.

Extraction of polyphenols from sugar beet pulp with water resulted in about 30% higher TPC compared to the extraction with methanol and acetone and almost 4 times higher yield than using 80% ethanol as an extractant (Table 3). However, it should be noted that the results obtained by the method using the Folin–Ciocalteu reagent, according to many authors [75], may be subject to error due to the fact that the above-mentioned reagent may react not only with polyphenols but also with other compounds such as amino acids and alkaloids, proteins, organic acids, polysaccharides, and vitamin C. Therefore, in this study, total polyphenol content (TPC) was also determined without using the Folin–Ciocalteu reagent by spectrophotometric method according to the methodology of Mazza et al. [55] with modification of Oomah et al. [56]. Methanol extraction was found to increase the level of determined phenolic compounds in the analyzed samples in the range of 7 to 22 times compared to ethanol and acetone extraction, respectively. It was noted that extracting the sample with water resulted in higher polyphenolic content than extracting with 80% methanol, ethanol, and acetone by 25%, 9.5-fold and 30-fold, respectively. Thus, it can be concluded that water is the best extractant and gives the highest polyphenol content and that the use of different organic reagents reduces the determined content of these components depending on the extractant used (Table 3). Considering the polyphenol content (TPC) determined according to the methodology of Mazza et al. [55] with modification of Oomah et al. [56], in the analyzed samples, it was found to be significantly lower (from about 8% to 26 times) than those obtained by the method of Singleton et al. [54] (Table 3), which should be explained as previously mentioned by the ability of the Folin–Ciocalteu reagent to form color complexes with other compounds besides polyphenols. Mohdaly et al. [40] studied the effect of various solvents, i.e., methanol, ethanol, acetone, hexane, and diethyl ether on the antioxidant properties of, among others, sugar beet pulp. According to the authors, methanol performed slightly better than ethanol as a solvent for phenolic compounds, flavonoids and flavonols, but the differences were minor. Therefore, for use in the food industry, the authors proposed ethanol as a more appropriate solvent. In the study of Boulekbache-Makhlouf et al. [42] on the polyphenol content of eggplant peel using three types of extractants (70% methanol, ethanol, and acetone), it was proved that 70% methanol was the best, while according to Sulaiman et al. [45], 70% acetone was found to be the most efficient solvent for polyphenolic compounds from vegetables. Do et al. [46] showed that the amount of polyphenols extracted from the test material was the highest after using ethanol and acetone extractants. In the study of Gawlik-Dziki and Kowalczyk [76], it was found that the content of total phenolic compounds extracted from radish sprouts was at the same level despite the use of different extractants. In the study of Makanjuola [77], which investigated the content of total polyphenols in an aqueous extract of ginger tea, it was clearly observed that water as an extractant was more beneficial to the isolation of total polyphenols than ethanol. In addition, in the case of our study, water proved to be the best extractant, followed by methanol, for both methods of polyphenol determination (with and without FCR). At this point it should be emphasized that the obtained results of the analysis of total polyphenols content, determined by the method both with and without a Folin reagent are promising. First of all, water was found to be the best extracting agent. This is advantageous as it indicates a wide range of potential

applications of water-based SBP extracts. They could be applied as products with potential health-promoting properties or as dietary supplements and pharmaceuticals. Furthermore, the nutritional value of the pulp would remain intact after being dried using this method.

The extraction of flavonoids from sugar beet pulp also revealed a significant effect of the extraction solvent. The highest flavonoid content was determined using 80% methanol as the extractant. This content was estimated to be as high as 370 mg of rutin per 100 g d.m. (Table 3). The least favorable effect on the flavonoid content of beet pulp was observed in the extraction with 80% ethanol and acetone, because when extracted with these reagents, their content was lower by 7 times and 2 times compared to methanol extraction, respectively (Table 3). In the study of Mokrani and Madani [44] on peach fruit, it was observed that the highest flavonoid content was recorded when these compounds were extracted in 60% acetone; similarly, in the study of Sulaiman et al. [45], 70% acetone extracted the highest amount of flavonoids from vegetables. On the other hand, in the study of Do et al. [46] on the effect of extractants on flavonoid content in the medicinal spice *Limnophilia aromatica*, high yields were noted using 75% methanol, 100% ethanol, and 75% and 100% acetone. For flavonoid content in oranges, it was observed that acetone is definitely a more favorable extractant than water [78]. On the other hand, Gawlik-Dziki and Kowalczyk [76], investigating the effect of flavonoid extraction conditions from radish sprouts, unequivocally found that both ethanol and methanol extraction lead to the same amount of flavonoids. In a study by Makanjuola [77] on flavonoid content in tea aqueous extract of ginger and aqueous extract of water and ginger, it was observed that ethanol was a better extractant than water.

Besides polyphenols and flavonoids, flavonols and phenolic acids were other very important bioactive compounds analyzed in this work, due to their significant health-promoting nature [79,80]. For phenolic acids, 80% methanol, acetone, ethanol, and finally water were found to be the most favorable extractants (Table 4). Considering the effect of the extraction solvent on the amount of flavonols in the sample, it was observed that the most favorable extraction was conducted in water and ethanol, while methane and acetone extraction oscillate at the same level (Table 4).

**Table 4.** The content of flavonols, phenolic acids, and antioxidant activity in extracts of freeze-dried sugar beet pulp, depending on the type of solvent used in the extraction process.

| Type of Polyphenol Extraction from SBP | Content of Phenolic Acids (mg FAE/100 g d.m.) | Content of Flavonols (mg QE/100 g d.m.) | TEAC (mM Trolox/kg d.m.) | FRAP (mM Fe$^{2+}$/kg d.m.) |
|---|---|---|---|---|
| methanol extraction | 250.67 ± 1.26 [d,*] | 95.86 ± 7.33 [a] | 38.15 ± 0.92 [d] | 71.75 ± 0.82 [c] |
| ethanol extraction | 13.40 ± 0.42 [b] | 143.07 ± 0.20 [b] | 17.37 ± 1.05 [b] | 64.41 ± 0.92 [b] |
| acetone extraction | 243.61 ± 0.50 [c] | 88.66 ± 8.50 [a] | 24.27 ± 1.93 [c] | 106.43 ± 1.67 [d] |
| water extraction | 3.42 ± 0.27 [a] | 202.00 ± 3.46 [c] | 9.35 ± 0.17 [a] | 48.88 ± 1.92 [a] |

* Different letters indicate significant differences between individual samples ($p \leq 0.05$); data are presented as mean ± SD.

Gawlik-Dziki and Kowalczyk [76], studying radish sprouts, proved that acetone extraction could be more effective for the determination of phenolic acids compared to methanol extraction. Moreover, mandarin fruit (*Citrus reticulate* L.) extracted using methanol showed a higher content of assayed phenolic acids in these extracts compared to ethanol extraction [81].

In the case of antioxidant activity determined through ABTS cation radical, it was noted that the samples after extraction with organic reagents, namely, methanol, ethanol, and acetone have higher activity than water extracts. This is most likely due to the fact that extraction of phenolic acids from samples could have a large effect on antioxidant activity. Ferulic acid (FA), constituting approximately 1% of dry weight of SBP, is a dominant phenolic acid [82] and shows a wide range of pro-health effects against cancer and diabetes, protects the cardiovascular system, and has a recognized potential for commercial applications in the food industry as a preservative due to its antioxidant and antimicrobial

activity, and as a crosslinking agent for gel formation [83,84]. A transparent example is water as the most favorable extractant of polyphenols and flavonols, but the worst of phenolic acids which resulted in the lowest antioxidant activity of the analyzed samples (Table 4). This is evidenced by the strong correlation ($R^2$ = 0.861) between phenolic acid content and antioxidant activity (method with ABTS), and the weak correlation between TPC with and without FCR with ABTS ($R^2$ = −0.125 and −0.096, respectively). A weak correlation was also noted between flavonoid content and ABTS ($R^2$ = 0.392) (Table 5).

**Table 5.** Correlation coefficients for the data.

|  | **ABTS** | **FRAP** |
|---|---|---|
| TPC with FCR | −0.125 | −0.092 |
| TPC without FCR | −0.096 | −0.709 |
| Flavonoids | 0.392 | −0.232 |
| Flavonols | −0.845 | −0.835 |
| Phenolic acids | 0.861 | 0.765 |

The analysis of the results of the antioxidant potential of the tested sugar beet pulp extracts, expressed by the ability to reduce iron (III) ions in the FRAP test (Table 4), showed that the extracts obtained using 80% aqueous acetone as the extractant were the most effective in this respect. In this case, the amount of reduced iron (III) ions was 106.43 mMol $Fe^{2+}$/kg d.m. Extracts obtained using water as an extractant showed two times lower activity in this respect, i.e., 48.88 mMol $Fe^{2+}$/kg d.m., compared to acetone extracts. On the other hand, the use of 80% water solutions of methanol and ethanol for the extraction of sugar beet pomace allowed obtaining extracts with the ability to reduce iron (III) ions, on average, 1.5 times lower compared to those obtained with the use of 80% acetone. Based on the results obtained using the FRAP test, it can be suggested that the ability to reduce iron (III) ions in the tested extracts is influenced by phenolic acids, not flavonols (Table 4). It is clearly visible that the extracts obtained with the use of 80% acetone and 80% methanol, containing the highest amount of phenolic acids, among the tested samples, were also characterized by a statistically significantly greater ability to reduce iron (III) ions compared to extracts obtained with solvents, i.e., 80% ethanol and water. This is also evidenced by the strong correlation between phenolic acid content and FRAP ($R^2$ = 0.765) and the strong inverse correlation between flavonols and FRAP ($R^2$ = −0.835) (Table 5).

## 4. Conclusions

It should be emphasized that sugar beet pulp after freeze-drying is a valuable source of polyphenols, especially phenolic acids, as well as other nutrients (protein, fat, sugars, and ash) and non-nutritive pro-health ingredients, i.e., dietary fiber (soluble and insoluble fractions). It has been shown that the main fractions of non-starch polysaccharides are cellulose, hemicellulose (arabinans), and pectins with specific spatial structure. Among the sugars building the polysaccharides, the most abundant are glucose, arabinose, rhamnose, and galacturonic acid, which makes sugar beet pulp an interesting raw material for further processing.

This study demonstrated the high content of arabinans in sugar beet pulp, indicating that it is a valuable raw material that could be utilized to obtain non-starch polysaccharides with potential bioactive properties due to the presence of phenolic acids in their structure. These polysaccharides could be used as carriers in encapsulation processes for bioactive compounds or drugs, leading to new controlled delivery systems that increase bioavailability and bioaccessibility. Moreover, the bioactive potential of these non-starchy polysaccharides could enhance the value of the resulting encapsulates. Additionally, arabinans isolated from sugar beet pulp could be used as a raw material to produce arabinose or its derivative, L-arabitol, which exhibit high health-promoting potential, including a low glycemic index and prebiotic properties. Therefore, sugar beet pulp has significant potential for applications in the food and pharmaceutical industries.

Moreover, it has been proven for the first time that water extraction was the most suitable extraction method for sugar beet pulp, because it allowed for determining the highest content of total polyphenols and flavonols. In the case of other groups of polyphenols, i.e., flavonoids and phenolic acids, as well as antioxidant activity, organic solvents (especially 80% methanol and acetone) proved to be the most efficient extractants for the samples analyzed. Based on this, it can be suggested that aqueous SBP extracts would be a low-cost, readily available intermediate product that could have applications as nutraceuticals or functional food additives.

Summarizing, sugar beet pulp is a source of ingredients suitable for pro-health functional food, which after microbiological stabilization, i.e., drying (e.g., freeze-drying) can be successfully used in food technology. This study revealed that sugar beet pulp contains not only a high concentration of phenolic compounds but also non-starch polysaccharides, specifically arabinans. These arabinans possess bioactive potential and contain arabinose in their structure, making them an attractive option as a carrier in the process of microencapsulation. The use of isolated arabinans from sugar beet pulp as a carrier in the encapsulation process has the potential to create value-added microcapsules.

**Author Contributions:** Conceptualization, A.B., D.G. and R.Z.; methodology, A.B., D.G., R.Z., J.R.-K. and K.M.; software, J.R.-K. and K.M.; validation, A.B., R.Z., D.G., J.R.-K. and K.M.; formal analysis, R.Z., D.G. and J.R.-K.; investigation, A.B., D.G., J.R.-K. and K.M.; resources, A.B. and J.R.-K.; data curation, D.G. and J.R.-K.; writing—original draft preparation, D.G., R.Z. and J.R.-K.; writing—review and editing, D.G., R.Z. and J.R.-K.; visualization, R.Z.; supervision, D.G. and J.R.-K.; project administration, A.B. and J.R.-K. All authors have read and agreed to the published version of the manuscript.

**Funding:** This research was financially supported by the Ministry of Science and Higher Education of Republic of Poland.

**Institutional Review Board Statement:** Not applicable.

**Data Availability Statement:** Not applicable.

**Conflicts of Interest:** The authors declare no conflict of interest.

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
