# Peer review of "Physicochemical Properties and Evaluation of Antioxidant Potential of Sugar Beet Pulp—Preliminary Analysis for Further Use (Future Prospects)"

_agriculture, doi:10.3390/agriculture13051039_

Round 1

Reviewer 1 Report

This paper is covering an interesting topic and the results contribute to the knowledge of the field. However, it contains various mistakes, some very serious, including missing references and incorrectly citing them.

*Introduction*

Lines 36-39- Reference is missing. Also, update data.

Lines 51 and 52- References 2 and 3 do not correspond to the stated information. 

Line 72- phenolic compounds are not necessarily “an integral part of fiber”- only some fibers have bounded phenolic compounds.

 Lines 79-84. References 14-18 do not fully support the stated health benefits. Also, the referenced papers are covering information mainly on flavonoids and flavones on cancer and glycemic effect.

Lines 89-91. Reference is missing

Line 137-140- This objective does not is in line with that stated in the abstract (lines 19-22).

*Materials and  methods*

Lines 157-158: The referenced method does not correspond to total carbohydrate content, it refers to total sugar content 

Line 184- monosaccharides composition?

Lines 210-255. It seems that Kazemi proposed only 3 of the total included formulas. It´s true?

Lines 257-265- how those conditions were selected?

Line 288- this reference is not the correct one. El Hariri referred to Andary 1990.

Lines 295 vs 306- antioxidant capacity or activity?

Lines 278-279:  this reference is not the original

*Results and discussion*

Line 336- this data is not presented in any of the results tables of this author. For example, Dinand informed a 7% of protein, not 5-5.5%.

Lines 351-353: reference is missing

Tables

Table 1- include the units of the values

For all tables, a space must be inserted between  the values and the symbol ±

Conclusions

Nothing is referred to as “safe” as stated in the title of this paper- same comments for results and discussion section

In general, the meaning and assays to support “safe” is not covered in a clear way along the text.

Reviewer 2 Report

I find this study valuable because sugar beet pulp is evaluated by technologically and academically professional individuals working in this field. It is important to evaluate and present the findings belongs to sugar beet pulp from bioactive potential on health.

Introduction part is well evaluated, giving clear information. MM part is a little confusing. Please reorganize for make it basic and systematic. Same goes for the results and discussion part. Tables should be improved; understandable and clear abbreviations should be preferred. Also, in title, you put health side in expectation with ‘pro-health’. In results and discussion, only analyze results and extract effect has included. These results should be associate with the bioactive potential. Sugar beet pulp is evaluated completely by content and components. Also, these results would be more meaningful with more statistical evaluation (Linear correlation could be added). I recommend to improve the manuscript expressively and holistically for expressing results more effectively.

These advises could be taken in consider for improving manuscript:

Line 149: Please give the name of the farm and revise as agricultural farm (Kruszwica, Piecki, Poland) by city name and country name.

Lines 149-153: Please revise the sentence as: The study material consisted of sugar beet pulp obtained from Agricultural farm (Kruszwica, Piecki, Poland) which was freeze dried (−47 °C, 37 Pa, 24 h; Labconco FreeZone 6, USA) and ground by a laboratory grinder (Grindomix, GM 200, Haan, Germany) at 7000 rpm in two consecutive runs each lasting for 5 sec.

Line 167: Please add brand name, modal name, city name and country name for the HPLC.

Line 175: Please add a comma after ‘pore size’

Line 183: Please give the brand name of the standarts as (Brand name, city name and country name)

Line 262: 15 min

Line 275: Please prefer different abbreviation for ‘2.7.3. Determination of Total Polyphenol Content (TPC) (without reagent F-C), phenolic acids, flavonols’

Line 320: Please indicate with program used for statistical evaluation.

Lines 320-330: 10.11, 6.90  and 4.16

For Table 3:

Sample names could be given as:

Sugar beet pulp extracted with 80% methanol: ME (Methanol extraction)

Sugar beet pulp extracted with 80% ethanol: EE (Ethanol extraction)

Sugar beet pulp extracted with 80% acetone: AE (Acetone extraction)

Sugar beet pulp extracted with water: WE (Water extraction)

Analyse names:

TPC (mg CE/100 g dw) total phenolic content as mg catechin equivalent per 100 g dry weight

TFC (mg RE/100 g dw) total flavonoid content as mg rutin equivalent per 100 g dry weight

And please differentiate the TPC without Folin Reagent from TPC

Reviewer 3 Report

Dear authors I revised the characterization study on sugar beet pulp extract and you can find my comments below:

The study aimed to reveal the health potential compounds in the sugar beet pulp extract. The introduction part of the study made me feel that the manuscript investigated the details of the sugar beet pulp extract and the results of bioactive properties of SBP extract were linked with the potential health benefits. Unfortunately, the study seems very basic level for its phenolic content and antioxidant activity measurement section. The sugar characterisation were done in details but phenolic content measurements were at very beginner level. Nowadays, the spectrophotometric determination of TPC is just using for determinig the phenolic potential of the extract. If the title is "Sugar beet pulp as a source of ingredients for safe and pro-health functional food", the readers is expecting the detailed characterisation of phenolic compounds and linking those results with antioxidant potential of the samples. Besides this, additional bioactive property measurements should be added for this study . Moreover, the detailed carbohydrate characterization was not supported with the health related properties experimentally. The health related discussions were done only based on literature results. Therefore, the manuscript  to be published in Agriculture journal (Q1 in Agronomy, Q2 in Plant Sciences) needs more detailed characterizaton and strong relations between the components of sugar beet pulp and health related ingredients.

The other comments are listed below:

1. Abstract section:

Could you please emphasize the significant results of the study? It seems an abstract of a review article.

2. Section 2.7.4. and 2.7.5. You do not need to give the details of the instrument again because it was detailed in previous section.  

Round 2

Reviewer 2 Report

I recommend acceptance of the manuscript.

Author Response

Dear Reviewer,

Thank you for taking the time and effort to review our manuscript. We appreciate your valuable input and feedback, which have greatly contributed to the final version of the article.

Reviewer 3 Report

Dear authors,

I revised the revised manuscript and you can my comment below.

The title says "physicocemical properties of sugar beet pulp" but the detailed characterisation  studies are not involved in the text. The sugar characterisation of pulp is well known and it has lack of novelty in the literature. The other results are also at basic level and do not worth to be published in this quality journal. Therefore, unfortunately I regret to inform that I do feel this manuscript has lack of  novelty and scientific quality.

Author Response

Dear Reviewer,

We believe the current title reflects the contents, although some issues addressed in the manuscript only confirm earlier results. In order to underline the novelties we added new paragraphs in the Introduction and Conclusions, and significantly revised the rest of the text. We hope that the current version of the manuscript meets the journal's requirements and would be suitable for acceptance. Thank you very much for your time and effort to improve our work.